A secretory hexokinase plays an active role in the proliferation of Nosema bombycis

Huang Yukang 1
Zheng Shiyi 1
Mei Xionge 1
Yu Bin 1
Sun Bin 1
Li Boning 1
Wei Junhong 1 2
Chen Jie 1 2
Li Tian 1 2
Pan Guoqing 1 2
Zhou Zeyang zyzhou@swu.edu.cn 1 2 3
Li Chunfeng licf@swu.edu.cn 1 2
1 State Key Laboratory Of Silkworm Genome Biology, Southwest University , Chongqing , Chongqing , China
2 Southwest University, Key Laboratory for Sericulture Functional Genomics and Biotechnology of Agricultural Ministry , Chongqing , Chongqing , China
3 Chongqing Normal University, College of Life Sciences , Chongqing , Chongqing , China
Silva Pedro
Electronic publication date: 2018 Sep 21
Publication date: 2018
Volume: 6
Electronic Location ID: e5658
Received 2018 Jun 8; Accepted 2018 Aug 28
Copyright: ©2018 Huang et al.
Copyright year: 2018
Copyright holder: Huang et al.
License: This is an open access article distributed under the terms of the Creative Commons Attribution License, which permits unrestricted use, distribution, reproduction and adaptation in any medium and for any purpose provided that it is properly attributed. For attribution, the original author(s), title, publication source (PeerJ) and either DOI or URL of the article must be cited.
License URL: https://creativecommons.org/licenses/by/4.0/

Keywords: Microsporidia, Hexokinase, Secretory protein, RNAi

Funding: National Natural Science Foundation of China 31472151 31702185 Fundamental Research Funds for the Central Universities XDJK2015A010 XDJK2018AA001 This work was supported by the National Natural Science Foundation of China (Grant number: 31472151, 31702185) and Fundamental Research Funds for the Central Universities (XDJK2015A010, XDJK2018AA001). The funders had no role in study design, data collection and analysis, decision to publish, or preparation of the manuscript.

==============================
The microsporidian Nosema bombycis is an obligate intracellular parasite of Bombyx mori, that lost its intact tricarboxylic acid cycle and mitochondria during evolution but retained its intact glycolysis pathway. N. bombycis hexokinase (NbHK) is not only a rate-limiting enzyme of glycolysis but also a secretory protein. Indirect immunofluorescence assays and recombinant HK overexpressed in BmN cells showed that NbHK localized in the nucleus and cytoplasm of host cell during the meront stage. When N. bombycis matured, NbHK tended to concentrate at the nuclei of host cells. Furthermore, the transcriptional profile of NbHK implied it functioned during N. bombycis’ proliferation stages. A knock-down of NbHK effectively suppressed the proliferation of N. bombycis indicating that NbHK is an important protein for parasite to control its host.

Introduction

Microsporidia are unicellular eukaryotes and obligate intracellular parasites that have broad host range (Cali & Takvorian, 1999). Microsporidia can infect invertebrates and vertebrates including human beings (Cali & Takvorian, 1999; Franzen, 2004). The life cycle of a microsporidia can be divided into the dormant extracellular and intracellular proliferation stages (Bigliardi & Sacchi, 2001). As obligate parasites, most microsporidia no longer have an intact tricarboxylic acid cycle or mitochondria, which implies an acute dependence on host energy during the intracellular stage (Hacker et al., 2014). In Encephalitozoon cuniculi and Trachipleistophora hominis, a series of nucleotide transport proteins (NTTs) can steal ATP from hosts (Heinz et al., 2014; Tsaousis et al., 2008). Owing to the absence of oxidative phosphorylation in microsporidia, glucose metabolism by glycolysis releases 7% of full ATP potential (Berg, Tymoczko & Stryer, 2007). Even so, some microsporidia still maintain their intact glycolytic pathways, which may function during the extracellular stage (Wiredu et al., 2017). Although some microsporidia, such as Enterocytozoon bieneusi, lack its intact glycolytic pathway, most microsporidia still possess a hexokinase (HK) (Wiredu et al., 2017), which implies that HK may be a necessary protein for the infection and proliferation of microsporidia.

Hexokinase is a multifunctional protein that can act in transcription regulation (Herrero et al., 1995; Niederacher & Entian, 1991) and apoptosis (Bryson et al., 2002; Gottlob et al., 2001), as well as with the cytokine neuroleukin (a nerve growth factor) (Chaput et al., 1988; Faik et al., 1988; Gurney et al., 1986). Furthermore, HK also participates in interaction between pathogen-host. For example, in macrophage, HK is an innate immune receptor for the detection of bacterial peptidoglycan (Wolf et al., 2016). In intracellular pathogen, such as Plasmodium falciparum, HK usually plays a basic role in the conversion of glucose to glucose 6-phosphate (Faik et al., 1988). A pathogen secreted HK was first reported in the microsporidia Nematocida parisii (Cuomo et al., 2012), which was also demonstrated using biochemical experiments (Reinke et al., 2017). This phenomenon was reported in Paranosema (Antonospora) locustae as well. The P. locustae HK was predicted to contain a signal peptide, and the indirect immunofluorescent assay (IFA) showed a nuclear localization in the host (Senderskiy et al., 2014; Timofeev et al., 2017). Then, special secreting hexokinase function which work as a regulator to increase ATP generation to parasite surface, was explored in Trachipleistophora hominis (Ferguson & Lucocq, 2018).

Nosema bombycis can infect Lepidoptera, including the economical insect silkworm, causing pebrine disease by vertical and horizontal transmission (Han & Watanabe, 1988). During infection, N. bombycis synthesizes many kinds of secretory proteins to control the host cells (Tian, 2013). Just like P. locustae, N. bombycis has an intact glycolysis pathway (Wiredu et al., 2017). Therefore, we hypothesized that NbHK can be secreted into host cells and has a significant impact on N. bombycis’ proliferation.

Materials & Methods

Hexokinase sequence analysis and ORF amplification

The amino acid sequence of NbHK was submitted to SignalP 4.1 Server (http://www.cbs.dtu.dk/services/SignalP/) and NCBI (https://www.ncbi.nlm.nih.gov/) for signal peptide and domain predictions. The HK of N. bombycis (GenBank Accession No. EOB11276.1) was amplified from genomic DNA (gDNA) by PCR. The amplification reaction consisted of 30 cycles of 94 °C for 15 s, 55 °C for 30 s, and 72 °C for 1 min using the forward primer 5′-CGCGGATCCATGATAATTTTCTATTGT-3′, containing a Bam HI restriction site (GGATCC), and the reverse primer 5′-CCGCTCGAGATAAATAATTCGATGTAAAG-3′ containing a Xho I restriction site (CTCGAG). PCR products were recovered (Omega, Norcross, GA, USA), integrated into pET-32 vector (TaKaRa, Kusatsu, Japan), then the recombinant vectors were transformed into Escherichia coli DH5α competent cells. The identified pET-32a-NbHK vector was sequenced by Sangon (Shanghai, China).

Protein expression, purification and polyclonal antibody preparation

The pET-32a-NbHK vector was transformed into E. coli Rosetta for expression. After cultivation in 37 °C, the recombinant bacteria was induced for 20 h at 16 °C with 1 mM IPTG in LB medium. Nickel chelating affinity chromatography (Roche, Basel, Switzerland) was used to purify the target protein fused with hexahistine. BALB/c mice were used to generate antiserum by immunizing with 100 µg recombinant HK protein homogenized with Freund’s adjuvant (V/V=1:1, Sigma-Aldrich, St. Louis, MO, USA) four times. Complete Freund’s adjuvant was used in the first injection, then the incomplete Freund’s adjuvant was used in the following injections. All animal experiments were approved by Laboratory Animals Ethics Review Committee of Southwest University guidelines (Chongqing, China) (Permit Number: SYXK- 2017-0019).

Indirect immunoinfluscent assay

Infected cells and healthy cells were fixed in PBS with 4% paraformaldehyde, then treated with 0.5% Triton X-100 for 10 min at room temperature, blocked with blocking regent (0.5% (v/v) bovine serum albumin and 10% (w/v) goat serum) for 1 h, incubated with 1:500 dilutions of HK-antiserum (mouse) and Nbβ-tubulin polyclonal antibody (rabbit) at 37 °C for 60 min. After washing with PBS three times (5 min each time), Alexa Fluor 488 and 594 (Thermo Fisher, Santa Clara, CA, USA) were used to detect the binding of primary antibodies at room temperature. The nuclei were stained with 4′, 6-diamidino-2-phenylindole (DAPI) (Thermo Fisher, Santa Clara, CA, USA) for 10 min. Samples were observed and imaged with Olympus FV1200 (Olympus, Tokyo, Japan).

Protein preparation

Protein of infected cells, mature spores and healthy cells were prepared by the glass-bead breaking method as reported (Geng et al., 2013). The samples with 0.4 g glass beads (212–300 µm), were lysed in RIPA Lysis Buffer (Beyotime, Shanghai, China) containing a protease inhibitor (phenylmethylsulfonyl fluoride), and then crushed for 5 min at 4 °C using a Bioprep-24 Homogenizer (ALLSHENG, Hangzhou, China). The treated samples were centrifuged at 12,000 rpm for 5 min and the supernatant were collected.

Immunoblot analysis

Proteins were separated by sodium dodecyl SDS-PAGE and transferred to polyvinylidene fluoride (PVDF) membrane (Roche, Basel, Switzerland). After blocking in blocking buffer (5% (w/v) skim milk, 20 mM Tris–HCl, 150 mM NaCl and 0.05% Tween-20), membrane incubated with an HK-antiserum or Nbβ-tubulin-antiserum (diluted 1:3,000) (Chen et al., 2017), washed, and incubated with HRP-labeled goat anti-mouse IgG (diluted 1:6,000; Sigma, Saint Louis, MI, USA). The bound antibodies were detected by ECL Plus Western Blotting Detection Reagents (Bio-Rad, Richmond, CA, USA). The protein concentrations were detected with BCA Kit (Beyotime, Shanghai, China), and loading quantity of samples were normalized on the basis of Nbβ-tubulin quantity.

Expression of recombinant HK fused with DsRed in BmN

NbHK was cloned using the forward primer 5′-GGGTACCATGATAATTTTCTATTGT CTAC-3′ or 5′-GGTACCTTAATTAAGACATTGGGAAATA-3′ (to remove the signal peptide (ΔSG)), each containing a KpnI restriction site (GGTACC) and the reverse primer 5′-TGACCCTGAGCCTCCATAAATAATTCGATGTAAAG-3′ containing a G3S2 linker sequence (GGAGGCTCAGGGTCA). The DsRed, a red fluorescent protein, was cloned used the forward primers 5′-GGAGGCTCAGGGTCAATGGTGCGCTCCTCCAAGAAC-3′ containing a G3S2 linker sequence (GGAGGCTCAGGGTCA) and the reverse primer 5′-CCTCGAGGCGGCCGCTACAGGAACAGG-3′ containing a Xho I restriction site (CTCGAG). Then the DsRed was linked to the rHK and rHKΔSG respectively through a G3S2 linker peptide by overlapping PCR. Then the overlapping PCR products were integrated into pCR-Blunt II-TOPO vector (Thermo Fisher, Santa Clara, CA, USA), after which the products of linkage were transform into E. coli DH5α. The recombinant pCR-Blunt II-TOPO vector contained the targets fragments, were extracted from E. coli DH5α. The above vectors and pIZ/V5-His (Thermo Fisher, Santa Clara, CA, USA) were digested by KpnI and Xho I (Thermo Fisher, CA, USA). The digested rHK and rHKΔSG fusing with DsRed were integrated into digested pIZ/V5-His, which was induced by T4 DNA ligase (New England Biolabs, MA, USA). Two µg constructed vectors were transiently transfected into BmN cells using X-tremeGENE™ HP DNA Transfection Reagent (Thermo Fisher, Santa Clara, CA, USA). Three days later, the transfected cells nuclei were labeled with Hoechest (Thermo Fisher, Santa Clara, CA, USA). Then the samples were examined by confocal microscopy (Olympus, Tokyo, Japan).

RNA interference (RNAi) fragments synthesis

The sequence of NbHK was submitted to BLOCK-iT™ RNAi Designer (http://rnaidesigner.thermofisher.com/rnaiexpress/design.do). A 352-bp fragment that contain five potential interferential dsRNA fragments was amplified by the primers F-RI-Hexokinase-T7 5′-TAATACGACTCACTATAGGGAGAAGGAATATACTTGTCTGGGA-3′ and R-RI-Hexokinase-T7 5′-TAATACGACTCACTATAGGGAGATTGACAGGTCTCT CAAATGC-3′, each containing the T7 promoters (TAATACGACTCACTATAGGGAGA). The amplified product was used as template to synthesize dsRNA using a RiboMAX Large Scale System-T7 Kit (Promega, Madison, WI, USA). The dsRNA-EGFP, which was used as the mock group, was prepared with the same method by the primers F-RI-EGFP 5′-TAATACGACTCACTATAGGGAGAACGGCAAGCTGACCCTGAA-3′ and R-RI-EGFP 5′-TAATACGACTCACTATAGGGAGATGTTGTAGTTGTACTCCAG-3′, each containing the T7 promoters as well.

DNA and cDNA collection from infected samples

Spores were separated from hemolymph of severely infected silkworm pupae. The spores which were pretreated with 0.1 mol/L KOH, were added to the Sf9-III cells (cell: spores ratio, 1:5) (Kawarabata & Ren, 1984). Infected cells were collected at 1, 3, 5 days post infection (d. p. i) and stored in PBS or TRIzol (Invitrogen, Carlsbad, CA, USA). The newly molted 4th instar silkworm larvae were oral fed with 1 × 107 spores per. Then, the infected silkworms’ midguts were collected and stored in TRIzol at −80 °C immediately. The gDNA of infected Sf9-III cells was extracted using a DNA Extraction Kit (Omega, Norcross, GA, USA), while the cDNA of infected cells and midguts were prepared using Total RNA Kit II (Omega, Norcross, GA, USA) and RT-PCR Kit (Promega, Madison, WI, USA). Samples were taken from three separate experimental, mock and blank groups at each time point.

RNAi of N. bombycis in infected Sf9-III

The Sf9-III (Thermo Fisher, Santa Clara, CA, USA) cells were infected by using the previous method described above. After infection, two µg dsRNAs of NbHK or EGFP were transfected into Sf9-III respectively and then cultured in 6-well plates (Saleh et al., 2016). The samples were collected at 1, 3 and 5 d. p. i.

Cell counting

The cell samples were suspended with one mL culture medium, then 10 µL cell suspensions were mixed with 10 µL trypan blue. Living counting were finished with Countess II FL (Thermo Fisher, Santa Clara, CA, USA).

Real-time quantitative PCR analysis

For transcription detections, one µg RNA was used to conduct reverse transcript PCR. Then, the products were diluted ten times as templates. Quantitative PCR was amplified using Hexokinase-qF 5′-CAAAATGTGATTATTATGGGAGATG-3′ and Hexokinase-qR 5′-CGATGTAAAGTATAAAGGGCTGAT-3′ primers, and reference gene primers SSU-qF 5′-CTGGGGATAGTATGATCGCAAGA-3′ and SSU-qR 5′-CACAGCATCCATTGGAAA CG-3′ (Huang et al., 2018). Quantitative PCR was performed with following program: a pre-denaturation of 95 °C for 2 min, followed by 40 cycles at 95 °C for 10 s and 60 °C for 20 s (LightCycle 96, Roche, Switzerland). Expression of SfHk was detected using the same method with SfHK-qF 5′-TCACTTACATTCAAGATTTACCCAA-3′ and SfHK-qR 5′-CTACGCCAGAACAAGAAAAGC-3′ primers, and reference gene primers SfGAPDH-qF 5′-GGCTGGCGCTGAATACATCGTTGAGTCCAC-3′ and SfGAPDH-qR 5′-TTAGCAACGGGAACACGGAAAGCCATACCAG-3′.

For Nbβ-tubulin copy number detection, one µL (∼200 ng/µL) of the each gDNA samples extracted from the above infected cells was analyzed by qPCR. The 10 µL reaction systems were conducted using the Nbβ-tubulin-qF 5′-AGAACCAGGAACAATGGACG-3′ and Nbβ-tubulin-qR 5′-AGCCCAATTATTACCAGCACC-3′ primers. Real-time PCRs were performed with the above program (LightCycle 96; Roche, Basel, Switzerland). The standard template had been constructed in previous research (Huang et al., 2018). The standard curve covered four orders of magnitude for the starting quantity (1.3 × 103–106). The multiple T tests were conducted with GraphPad.Prism.v6.01.

Results

Recombinant hexokinase purification and immunoblot analysis

The amino acid sequence analysis showed that NbHK contained a signal peptide, which implied it was a secretory protein (Fig. 1A). The ORF of HK was cloned by PCR using specific primers based on the gDNA of N. bombycis. The target gene were successfully integrated into pET-32a vector, which was validated by PCR and restriction enzyme digestion (Fig. 1B). The sequencing results showed a 1,287-bp fragment encoding 428 amino acids, which was consistent with the data from the genomic sequence of N. bombycis in SilkPathDB (https://silkpathdb.swu.edu.cn/) (Fig. S1).

Figure 1 Expression and analysis of NbHK.

(A) Protein structure of NbHK. There is a signal peptide contacting 23 amino acids in the N terminus. (B) Validation of the pET-32-NbHK vector by PCR (left) and Bam HI/Xho I enzyme digestion (right). Products 1,300 bp were amplified by PCR or cleaved from the constructed vector. M: 2K Plus (TaKaRa, Kusatsu, Japan). (C) Purification of recombinant HK. Recombinant HK was eluted by elution buffer containing different concentration of imidazole and analyzed by SDS-PAGE. (D) Specificity of the HK antiserum. Proteins extracted from infected Sf9-III, mature spores and healthy Sf9-III were subjected to western blot using polyclonal antibody against HK. M, Protein maker (Transgene, Shanghai, China).

SDS-PAGE analysis showed that rHK was expressed at a molecular mass of ∼60 kDa, which was consistent with the predicted size (Fig. 1C). The purified protein was used to prepare the antibody. The titer of the HK antiserum was determined by ELISA (Fig. S2). Western blot indicated that the HK antiserum can recognize a 50 kDa protein specifically in the infected cells proteins, but the signal was not demonstrated in mature spores and healthy Sf9-III cells. (Fig. 1D). The molecular mass difference between rHK and native HK, which was consist in infected cells, was due to tag proteins.

Subcellular localization of NbHK

The subcellular localization of NbHK was analyzed by IFA. Nbβ-tubulin was used as a marker protein to distinguish the meront stage from mature spores. Meronts can be labeled using the Nbβ-tubulin antibody, while the mature spores can be observed visibly in differential-interference microscopy (DIC) (Chen et al., 2017; Huang et al., 2018). The IFA demonstrated the NbHK can be secreted into host cells (Fig. 2). In the proliferation stage, NbHK was located in the nucleus and cytoplasm of host cell (Figs. 2A–2E). When N. bombycis matured, the NbHK tended to be concentrated in the nuclei of host cells (Figs. 2F–2J). The change in localization implied that NbHK’s functions might be different, depending on the growth stage of N. bombycis. The antibody of NbHK cannot bind to host cell in uninfected cells (Fig. S3).

Figure 2 Subcellular localization of NbHK in infected Sf9-III.

Localization of NbHK was imaged with confocal microscopy at different stage of infection. Green and red fluorescence were observed in the samples incubated with the HK antibody (mouse) and rabbit Nbβ-tubulin antiserum, respectively. The nuclei of host cells and N. bombycis were labeled with DAPI (blue fluorescent signal) (Thermo Fisher, Santa Clara, CA, USA). (A–E) Localization of NbHK in the meront stage of N. bombycis. The parasites were bound by Nbβ-tubulin antibody, which showed that the meronts lacked a chitin layer. The NbHK dispersed in the whole host cell. (F–J) Localization of NbHK in the mature spore stage of N. bombycis. The chitin layer of the mature spore blocked the binding of the Nbβ-tubulin antibody, and the mature spore could be observed by DIC. NbHK was concentrated in host cell nuclei at this stage.

Figure 3 Subcellular localization of HK::DsRed and HKΔSG::DsRed in BmN.

(A) HK, with or without its signal peptide, was fused with DsRed. The two expression cassettes were integrated into pIZ/v5-His, and then the constructed vectors were transfected into BmN. (B) Subcellular localization of HK::DsRed in BmN. The cell nucleus was labeled with Hoechest. The white arrow indicates an apoptotic cell. (C) Subcellular localization of HKΔSG::DsRed in BmN.

The vectors that encode fusion proteins HK::DsRed and HKΔSG::DsRed were independently transfected into BmN cells (Fig. 3A). The fluorescence signal showed that rHK was localized in the cytoplasm of healthy cells, independent of presence of its signal peptide (Figs. 3B, 3C). When the BmN cells’ nuclei became diffuse, indicating a trend toward apoptosis, the HK::DsRed co-localized with the nuclei (Fig. 3B, white arrow). The similar results were demonstrated in transfected Sf9-III cells (Fig. S4).

Transcriptional profile of NbHK in infected cells and midguts

In infected Sf9-III cells, NbHK was expressed at the early stage of infection and showed a highly significant increase from 8 to 48 h post infection (h. p. i). Then, transcription levels were relatively stayed stable from 48 to 72 h.p.i and were slightly down-regulated from 72 to 96 h.p.i (Fig. 4A). In infected midgut of B. mori, the NbHK expression pattern was similar to that of infected cells. NbHK was gradually up-regulated, except for a slight down-regulation at 24 h. p. i (Fig. 4B). The transcript profile of NbHK implied it functioned during all proliferation stages of N. bombycis.

Figure 4 Transcriptional profile of NbHK in infected Sf9-III cells and midguts.

The relative expression level of NbHK in infected Sf9-III cells and midguts at different time points. They are presented relative to the 8 h.p.i. N. bombycis SSU rRNA was the reference gene for the normalization of expression levels. (A) Transcript level of NbHK in infected Sf9-III cells. (B) Transcript level of NbHK in infected midgut of B. mori. Vertical bars show the mean ± SEs (n = 3).

Down-regulation of NbHK suppressed N. bombycis’ proliferation

The effects of RNAi were assessed using qPCR and western blot. Transcriptional levels revealed that the expression of NbHK was significantly down-regulated by dsRNA in the experimental groups (dsRNA-HK) compared with the mock groups (dsRNA-EGFP) at three and five d.p.i (Fig. 5A). Besides, the western blot also showed NbHK down-regulated at three d.p.i, which was basically consistent with transcriptional detection result (Fig. 5B).

Figure 5 Effect of NbHK down-regulation on N. bombycis proliferation.

(A) The transcription levels of NbHK. Complementary DNAs of the experimental (dsRNA-NbHK) and mock (dsRNA-EGFP) groups were analyzed by qPCR. The △△Ct method was conducted to process the data. (B) NbHK expression levels of NbHK. Total proteins of experimental (dsRNA-NbHK) and mock (dsRNA-EGFP) groups were analyzed by western blot. Loading quantity of samples were normalized by Nbβ-tubulin, and NbHK quantities were detected with its antiserum. (C) Infection levels of Sf9-III. Genomic DNA was extracted from experimental (dsRNA-NbHK) and mock (dsRNA-EGFP) groups at 1, 3 and 5 d.p.i. Copy numbers of Nbβ-tubulin indicated the infection levels. Statistically significant differences are represented with asterisks (*P < 0.05; ** P < 0.01). Bars represent the standard deviations of three independent replicates.

The copy number of Nbβ-tubulin which lacks homology in the host was used to the reflect the infection level of the two groups (Huang et al., 2018). The infection of level the two groups kept similar at 1 d.p.i, because the effect of RNAi was not obvious at this time. In the mock groups, N. bombycis began to proliferate from one to five d.p.i while the proliferation of N. bombycis was remarkably inhibited in the experimental groups at three d.p.i (Fig. 5C).

The dsRNA-EGFP, dsRNA-HK and transfection have no effect on proliferation of host cells and expression of SfHK (Figs, S5A, SB). Besides, the RNAi constructs have no impact on proliferation of N. bombycis (Fig. S6).

Discussion

NbHK was predicted to be a secretory protein with a signal peptide, suggesting its potential localization inside host cells. The location of NbHK during the proliferation stage of N. bombycis infection implied it may play a role in regulating and controlling the host’s energy metabolism (Ferguson & Lucocq, 2018). HK, with or without its signal peptide, and fused with DsRed was expressed in BmN cells, which showed a similar subcellular localization. The localization of eukaryotic expressed rHK revealed that the signal peptide could not lead the rHK to be secreted or to enter into nuclei of healthy BmN or Sf9-III. The localization results were different from those of P. locustae HK which is mainly concentrated on hosts’ nuclei (Senderskiy et al., 2014; Timofeev et al., 2017). Expression profiles in vivo and in vitro also indicated NbHK continuous high transcriptional level during proliferation stages of N. bombycis infection and the western blot showed NbHK barely exist in mature spores, which implied the NbHK just functioned in intracellular stage. But how microsporidia kept basic energy metabolism was still a secret. It is uncommon for an energy metabolism-related enzyme to be secreted into host cells and acted as a potential moderator. Although we showed that the down-regulation of NbHK expression can inhibit the proliferation of N. bombycis, the specific mechanisms involved need further study.

In microsporidia, there is only a tiny mitochondrial remnant called a mitosome (Goldberg et al., 2008; Williams et al., 2002), which implied that they are less prone to produce energy by themselves. The unique NTTs of microsporidia allow for a simplified energy metabolism pathway (Dean et al., 2018; Tsaousis et al., 2008). Although there is no ATP generation during the conversion of glucose to glucose 6-phosphate, it indirectly promotes glycolysis. The high transcription of NbHK and its importance function implied the glycolysis also play a key role during its intracellular stages, even microsporidia could steal ATP from host. HK is a control valve of glycolysis. The secretory HK of microsporidia turn up the valve of glycolysis and the tricarboxylic acid cycle, which is a clever strategy to steal energy from the host (Ferguson & Lucocq, 2018).

The lack of a stable and reliable gene manipulation method has impeded research on N. bombycis protein functions. In 2016, an EGFP was expressed in N. bombycis through a non-transposon vector (Rui et al., 2016). This was the first step toward the generation of successful transgenic microsporidia, but there were no follow-up reports regarding its application in protein function research. RNAi has been widely applied in other fungi including Nosema ceranae and Heterosporis saurida (Paldi et al., 2010; Saleh et al., 2016), to study protein functions. N. bombycis has all of the genes required for RNA silencing and it functions in vivo, but the RNA interference efficiency was unstable in different experimental groups (Pan et al., 2017; Wang et al., 2015). In our research, a stable RNAi strategy was established through an in vitro transcription system and lipofection in infected Sf9-III, which allowed high-throughput targets screening. Furthermore, RNAi in N. bombycis is not only a research method, but also an important strategy for breeding N. bombycis-resistant B. mori, which have been successful in BmNPV resistance (Kanginakudru et al., 2007).

Conclusion

The N. bombycis hexokinase is a secretory protein which localized in host cells’ cytoplasm and nuclei. The NbHK expressed during proliferation stage of infection in vivo and in vitro. The down-regulation of NbHK could inhibit the proliferation of this parasite. The results implied the NbHK was involved in controlling of host’s energy metabolism.

Supplemental Information

Figure S1 Nucleotide and amino acid sequences of NbHK

The NbHK sequencing results showed a 1,287-bp fragment encoding 428 amino acids.

Click here for additional data file.

Figure S2 Measurement of HK antiserum titer

The titer of HK antiserum was detected by ELISA. Titer of unimmunized BALB/c mouse was as the negative control. The result of ELISA showed the titer of HK antibody was 1:102400.

Click here for additional data file.

Figure S3 Negative control of group in IFA

Healthy Sf9-III cells were used to detect specificity of Nbβ-tubulin and NbHK antiserum. Cell nuclei were label with DAPI (blue). Binding of Nbβ-tubulin and NbHK were detect with Alexa 594 and 488 respectively. There were no any signals of Nbβ-tubulin and NbHK in healthy Sf9-III cells ’nuclei and cytoplasm.

Click here for additional data file.

Figure S4 Subcellular localization of HK::DsRed and HK △SG::DsRed in Sf9-III

HK with and without signal peptide fusing DsRed were located in cytoplasm.

Click here for additional data file.

Figure S5 The impact of RNAi constructs on host cells

Quantity PCR and live cell counting were used to detect the impact of dsRNA-EGFP or dsRNA-NbHK on host cells. Equal numbers of Sf9-III cells were transiently transfected with dsRNA-EGFP or dsRNA-NbHK. Blank group was without any dsRNA. (A) The impact of RNAi constructs on cell proliferation. Ten microliters cell suspension samples were mixed isopycnic trypan blue, and then the living cells were detected by Cell Count. (B) The transcription levels of Sf9-III hexokinase. Complementary DNAs of the blank, experimental (dsRNA-NbHK) and mock (dsRNA-EGFP) groups were analyzed by qPCR. GAPDH of Sf9-III was reference gene to normalize samples. The △△Ct method was conduct to process the data. Vertical bars show the mean ±SEs (n = 3).

Click here for additional data file.

Figure S6 The impact of non-specific RNAi on N. bombycis proliferation

N. bombycis proliferation in Sf9-III cells. Genomic DNA was extracted from blank and mock (dsRNA-EGFP) groups at 1, 3 and 5 d.p.i. Copy numbers of Nbβ-tubulin indicated the proliferation of N. bombycis. Vertical bars show the mean ± SEs (n = 3).

Click here for additional data file.

Table S1 HK transcriptional profile in vitro raw data

Real-time quantitative PCR raw data of transcriptional profile in vitro which include Cq of target and reference gene. HK was target gene. SSU was reference gene. ”h” means h. p. i. “d” means d. p. i.

Click here for additional data file.

Table S2 HK transcriptional profile in vivo raw data

Real-time quantitative PCR raw data of transcriptional profile in vivo which include Cq of target and reference gene.HK was target gene. SSU was reference gene. “h” means h. p. i. “d” means d. p. i.

Click here for additional data file.

Table S3 HK transcriptional level detection raw data in RNAi experiment

Real-time quantitative PCR raw data of HK transcriptional level in experimental and mock groups. which include Cq of target and reference gene. HK was target gene. The number behind the “HK” and ”EGFP” means d. p. i. EGFP was the mock group.

Click here for additional data file.

Table S4 β -Nbtubulin copy number detection raw data in RNAi experiment

Real-time absolute quantitative PCR raw data of β -Nbtubulin copy number detection which include Cq and concentration (copy number) of β -Nbtubulin in blank, experimental and mock groups.Blank 1, 3, 5d mean Blank groups at 1, 3, 5 d. p. i . DsRNA-EGFP 1, 3, 5d mean control groups at 1, 3, 5 d. p. i. DsRNA-NbHK 1, 3, 5d mean experimental groups at 1, 3, 5 d. p. i. Standards were the standard samples which copy number had been known.

Click here for additional data file.

Data S1 The raw image of Fig. 1

The full-length electrophoretic gels and western blot of Fig. 1.

Click here for additional data file.

Data S2 The raw image of Fig. 5B

The full-length western blot of Fig. 5B.

Click here for additional data file.

Data S3 Melting curve of qPCR

SYBR Green was used in all qPCR. Melting curve of NbHK, NbSSU, Nb β-tubulin, SfGAPDH and SfHK demonstrate the amplifications were specific.

Click here for additional data file.

Additional Information and Declarations

Competing Interests

Author Contributions

Animal Ethics

Data Availability

The authors declare there are no competing interests.

Yukang Huang conceived and designed the experiments, performed the experiments, analyzed the data, contributed reagents/materials/analysis tools, prepared figures and/or tables, authored or reviewed drafts of the paper, approved the final draft.

Shiyi Zheng performed the experiments.

Xionge Mei, Bin Sun and Boning Li conceived and designed the experiments.

Bin Yu contributed reagents/materials/analysis tools.

Junhong Wei, Guoqing Pan and Zeyang Zhou contributed reagents/materials/analysis tools, authored or reviewed drafts of the paper, approved the final draft.

Jie Chen and Chunfeng Li conceived and designed the experiments, contributed reagents/materials/analysis tools, authored or reviewed drafts of the paper, approved the final draft.

Tian Li conceived and designed the experiments, authored or reviewed drafts of the paper, approved the final draft.

The following information was supplied relating to ethical approvals (i.e., approving body and any reference numbers):

All animal experiments were conducted in accordance with Laboratory Animals Ethics Review Committee of Southwest University guidelines (Chongqing, China) (Permit Number: SYXK-2017-0019).

The following information was supplied regarding data availability:

The raw data are provided in the Supplemental Files.

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
