# Peer review of "A secretory hexokinase plays an active role in the proliferation of Nosema bombycis"

_PeerJ, doi:10.7717/peerj.5658_

## Round 0.1 · original submission · Major Revisions

Our reviewers are generally enthusiastic about your manuscript. Please provide all controls requested by reviewers #3 and #4 and perform the (relatively minor) tetual/graphical corrections highlighted in the other reviews.

·

Basic reporting

no comment

Experimental design

no comment

Validity of the findings

no comments

Additional comments

Huang and co-workers studied the location of hexokinase and its impacts on the spore proliferation. This is an interesting topic and the result could be used to control N. Bombycis infection. The experiment is well designed and the results are robust. I have only few comments:
1. Line 37: microsporidia can also infect healthy human. the sentence sounds odd.
2. Line 55: need reference
3. Line 67: consider replace “significant function” with “significant impact”
4. Line 90: use the full name
5. Line 122: has the off-target effect been controlled?
6. Line 143-145: it is confusing. infected samples mean cells or individual silkworm? the infection silkworms were immediately stored or later?
7. Line 204: has the off-target effect been controlled?
8. Line 209: revise “the infection of level the two groups”
9: Line 236: is there any reference?

·

Basic reporting

The text is clear, unambiguous and technically correct. However, I am not a native speaker and can not professionally evaluate it. The MS include sufficient references to demonstrate its actuality. The structure of the paper is correct and consists of standart sections. It is self-contained and all results are relevant to the research topic.

Experimental design

Submitted article represent original primary research within Aims and Scope of the journal. It clearly defines the research question and knowledge gap which the study has filled. The investigation was carried out with the use of technically advanced methods and with observance of ethical standards. The most part of methods are described with sufficient information to reproduce them. Some remarks are listed below.

Validity of the findings

The data are robust and statistically reliable. They are base for conclusion which is limited to supporting results, well stated and connected to the original research question.

Additional comments

Since 2012 (Cuomo et al. 2012) hexokinase has become one of the most interesting enzymes of microsporidia because of (1) its secretion into an infected cell and (2) accumulation in host nuclei.
This paper focuses on the hexokinase of microsporidia Nosema bombycis - a dangerous pathogen of silkworms, that emphasizes its practical significance. In this study, the authors performed heterologous expression of hexokinase in E. coli, produced immune serum against recombinant protein and for the first time showed that parasite enzyme is secreted into cytoplasm of infected cell and accumulated in host nuclei during microsporidia maturation. In addition, the authors showed NbHK expression at the early stage of infection and relatively stable transcription level during all Nb proliferation stages.
Very interesting result of this research is the suppression of Nb development by RNAi. It not only emphasizes the functional significance of hexokinase for the parasite, but also opens up new approaches for creating microsporidia-resistant insects.
At the same time I would like to make some remarks. I think, after consideration of the listed proposals, the article can be accepted for publication.

Lines 46, 47, 65 (Wiredu Boakye et al., 2017) or (Wiredu et al., 2017) is correct ref. ?

Line 56 Plasmodium instead of plasmodium

Line 83 “the recombinant bacterial was induced” or “the recombinant bacteria was induced”?

Line 86 Please note which (complete or incomplete) Freund's adjuvant was used?

Line 115 KpnI site is GGTACC instead of GGGTACC

Line 113 – 119 In contrast to the plasmid for bacterial expression (lines 72-80) the construction of vector for HK-G3S-RFP expression in insect cell is described not fully enough.
All information that can be obtained:
1. Forward primers as well as pIZ/V5-His MCS contain KpnI site.
2. 5’-end of Reverse primer tgaccctgagcctcc (ggaggctcagggtca) encodes GGSGS linker instead of G3S.
Please describe the strategy of chimeric gene construction . It would also be interesting to know the reagent and the effectiveness of lipofection, and whether zeocin was used to select the transformants.

Lines 170 -174 Since “The identified pET-32a-NbHK vector was sequenced by Sangon (Shanghai, China)” and “The sequencing results showed a 1,287-bp fragment encoding 428 amino acids, which was consistent with the data from the genomic sequence of N. bombycis in SilkPathDB (Fig 1 C)”, images of agarose gels may be omitted from the results - It rather refers to materials and methods.

Lines 173-179 in the text: “ which was consistent with the data from the genomic sequence of N. bombycis in SilkPathDB (https://silkpathdb.swu.edu.cn/) 174 (Fig 1 C). SDS-PAGE analysis showed that rHK was expressed at a molecular mass of ~60 kDa, which was consistent with the predicted size (Fig 1 D). The purified protein was used to prepare the antibody. The titer of the HK antiserum was determined by ELISA (Fig S1). Western blot 178 indicated that the HK antiserum can recognize a 50 kDa protein specifically in the infected cells 179 proteins (Fig 1 E).
Fig. 1 legend: “(C) Purification of recombinant HK. Recombinant HK was eluted by elution buffer containing different concentration of imidazole and analyzed by SDS-PAGE. (D) Specificity of the HK antiserum. Proteins extracted from infected Sf9-III were subjected to western blot using polyclonal antibody against HK. M: Protein maker (Transgene, China).”
(E ) - is absent!

Line 186 I do not like the phrase “in meront stage” It looks like localization in microsporidia meronts.

Line 202 “The transcript profile of NbHK implied it functioned during all proliferation stage” or “stages”?

Lines 215-216 “The location (cytoplasmic?) of NbHK during the early stage of N. bombycis infection implied it may play a role in regulating and controlling the host’s energy metabolism” .
Only localization of parasite enzyme in host cytoplasm is not sufficient reason to assume its participation in regulation of host metabolism. For example, protein may be accumulated in host cytoplasm simply because it is secreted into this compartment by microsporidia meronts and sporonts (in contrast to late stages and spores). At the same time, secretion of key enzyme of glycolysis (regardless of in which compartment it will accumulate) implies itself such important role.

Line 217-218 “while nuclear localization signal during the late stages of infection suggested it functions in apoptosis (Bryson et al. 2002; Gottlob et al. 2001).
In both of the cited articles there is no mention of the nuclear localization of hexokinase. I think, so far there is no link between the nuclear localization of hexokinase and apoptosis regulation.

Lines 223-224.
“The cytoplasmic localization of NbHK revealed that NbHK may regulate energy metabolism and the apoptosis of host mitochondria.”
Why localization of parasitic enzyme in host cytoplasm indicates the possibility of its participation in regulation of apoptosis of host mitochondria ?

Lines 230-231. Why Goldberg et al., 2008? First paper about microsporidial mitosomes - Williams BAP, Hirt RP, Lucocq JM & Embley TM (2002) A mitochondrial remnant in the microsporidian Trachipleistophora hominis. Nature 418: 865–869.

Line 232. “The unique NTTs of microsporidia allow for a simplified energy metabolism pathway (Dean et al. 2018).”
I would add here Tsaousis, A. D., et al. 2008. A novel route for ATP acquisition by the remnant mitochondria of Encephalitozoon cuniculi. Nature 453:553–556.

·

Basic reporting

See general comments

Experimental design

See general comments

Validity of the findings

See general comments

Additional comments

Microsporidia are a fascinating phylum of obligate intracellular pathogens that infect many animal species. They cause both death and disease in humans and agriculturally important animals and there is both fundamental and practical reasons for understanding how they can interact with their hosts. Microsporidia likely use many host-exposed proteins to interact with and reprogram hosts to a beneficial state for the pathogen. Here, the authors study a hexokinase from Nosema bombycis, one of the few proteins that microsporidia secrete with known molecular function. Using antibodies and heterologously expressed protein, the authors show the protein is localized to the nucleus and the cytoplasm in host cells. Impressively, the authors also provide a convincing example of using RNAi to knockdown hexokinase and demonstrate that it reduces the amount of the microsporidia. If this approach to RNAi is generalizable to other genes, this would provide a powerful approach to study N. bomybisis proteins and to potentially control the disease it causes in silkworms. There are several points to address before publication:

Major points
1. There needs to be data to demonstrate that the antibody used against hexokinase is specific in both western blots and microscopy. Line 177-179 mentions figure 1E, which doesn’t exist, but is stated to include western blot data where the band of the expected size is only observed in infected cells. This should be included, as well as microscopy data showing that the antibody doesn’t bind to uninfected cells.
2. For Figure 5 there are statistical tests performed on the RNAi treated samples, but no description of the test is provided. This needs to be included.
3. Lines 235-236 state that secreted hexokinases from microsporidia increase glycolysis in hosts. All though this is likely, to the best of my knowledge this has yet to be demonstrated. Also lines 253-254 state that hexokinase controls host energy metabolism. These statements need to be changed to indicated that although it is likely that hexokinases from microsporidia are involved in altering host metabolism, it is not yet known that this is the case and thus provides avenues for future research.

Minor points
1. Figure 4A would look better if presented as a log scale on the X-axis, rather than broken up as it is currently.
2. There is work on describing hexokinases from Nematocida microsporidia that would be helpful to cite. Namely Cuomo et al 2012, on describing that microsporidia have hexokinases with signal peptides, and Reinke et al 2017 demonstrating using biochemical experiments that hexokinases from two Nematocida species are observed in host cells.
3. The Y-axis units on Figure 5B could be changed from “starting quantity” to a more informative unit.

Reviewer 4 ·

Basic reporting

A reasonable study of the role of hexokinase in the microsporidia Nosema bombycis. The authors have used a standard approach of cloning the gene and immunolocalization to provide some insights into the potential role of this protein. However, they need to provide appropriate controls for the immunolocalization studies. In addition, they provide some data on RNAi in Nosema bombycis and this is important, as genetic manipulation of this group of pathogens has been difficult and has not previously been confirmed. To that end, it is critical that they carefully document the RNAi experiment to prove that they are inhibiting the gene of interest. The data are suggestive that the RNAi has worked but additional controls are needed.

The paper could used redaction to improve English syntax.

Overall, this is a reasonable paper, but it needs control data in the figures to accept the conclusions that were presented in the discussion.

Experimental design

The antibody was raised to Nb Hexokinase cloned using standard techniques. The data confirm expression of NbHx and the production of an antibody. What is missing is data on the cross reaction of this antiserum with uninfected host cells. Hexokinase is a relatively conserved gene and the antibody was raised to the Hx domain and thus is likely to cross react with other Hx. The figure should contain a lane for uninfected host cells as well as lane of purified spores of Nb. This would help confirm the specificity of the antiserum. In addition, the images on IFA need uninfected controls as well. It is important to demonstrate that the anti-NbHx serum does not react with the cytoplasm or nucleus of uninfected cells.

The RNAi experiment is interesting but more details are needed as the authors Line 245 to 247 indicate a system has been developed for Nb for this, but it is not referenced and a search of PubMed did not identify an article on this topic from this group. The experiment Fig 5A shows that all of the data is controlled for RNAi using EGFP (set to 1) and thus there is no error bar (if so then the graph of EGFP is meaningless as it is fixed at 1; The issue being does the control level vary over time cannot be obtained from the data as presented). It would be useful to show a control where EGFP is suppressed as well as a readout to demonstrate that the RNAi indeed works. The growth assay in FIg 5B needs to include growth in the absence of RNAi as a separate control (e.g. does RNAi nonspecifically suppress growth). Finally, if growth is reduced and the Hx level is lower is this due to the fewer numbers of organisms or to the RNAi. Data should be expressed as the RNA content for the gene per organism to demonstrate that the RNAi suppressed the transcript. Fig 5A has a problem in that if the EGFP increases in number then the relative transcript of Hx will be lower. Protein data would be useful to show the RNAi decreased the expression of Hx in the cells (or the amount of Hx found in host cells). Controls to demonstrate that the RNAi constructs are not having their effects to changes in the host cell (leading to less growth of the microsporidia) are also needed. (e.g. does the dsRNA suppress host cell Hx; what is the data that the microsporidian Hx is being suppressed). Overall, a very interesting and potentially exciting experiment, but more data is needed to make this compelling.

Validity of the findings

See section 2. The data needs more controls to clearly demonstrate the findings

Another issue if the comment that localization of Hx to the nucleus is due to apoptosis. There is not data presented to support this. There are many ways to confirm (by staining) that a cell is undergoing apoptosis and if the authors want to make this argument for the localization of Hx then they need to do cell staining to demonstrate that these cells have apoptotic markers.

Additional comments

Overall, there are some very interesting experiments presented and with appropriate controls this would be an exciting paper. The RNAi observation is extremely important due the absence of genetic techniques for these organisms and this needs to be controlled and presented so that it is evident that the RNAi technique is working on the microsporidia.

---

## Round 0.2 · accepted · Accept

Please address the final typos found by reviewer #2 with our production team.

# ·

Basic reporting

The text is clear, unambiguous and technically correct. However, I am not a native speaker and can not professionally evaluate it. The MS include sufficient references to demonstrate its actuality. The structure of the paper is correct and consists of standard sections. It is self-contained and all results are relevant to the research topic.

Experimental design

Submitted article represent original primary research within Aims and Scope of the journal. It clearly defines the research question and knowledge gap which the study has filled. The investigation was carried out with the use of technically advanced methods and with observance of ethical standards. The most part of methods are described with sufficient information to reproduce them. Some remarks are listed below.

Validity of the findings

The data are robust and statistically reliable. They are base for conclusion which is limited to supporting results, well stated and connected to the original research question.

Additional comments

In my opinion, the authors have considered and corrected all remarks and suggestions. Thus, the article can be accepted for publication. Please only check some minor mistakes like
L 105-106. “…cells were prepared by the glass-bead breaking method as reported” instead of “…cells were prepared by the glass-bead breaking method as report”
L 138-139. “X-tremeGENE™ HP DNA Transfection Reagent” instead of “X-tremeGENE HP DNA Transfecti Reagent”.
L 256. “(Ferguson & Lucocq 2018)..”

·

Basic reporting

See general comments

Experimental design

See general comments

Validity of the findings

See general comments

Additional comments

The authors have adequately addressed all of my concerns and I now enthusiastically support the publication of the article.